# Advancements in Anterior Cruciate Ligament Repair—Current State of the Art

Francesco Bosco [1,2,3], Giuseppe Rovere [4,5], Fortunato Giustra [6,*], Virginia Masoni [7], Salvatore Cassaro [3], Marcello Capella [7], Salvatore Risitano [7], Luigi Sabatini [8], Ludovico Lucenti [3] and Lawrence Camarda [3]

1. Department of Precision Medicine in Medical, Surgical and Critical Care (Me.Pre.C.C.), University of Palermo, 90133 Palermo, Italy; francesco.bosco03@unipa.it
2. Department of Orthopaedics and Traumatology, G.F. Ingrassia Hospital Unit, ASP 6, 90131 Palermo, Italy
3. Department of Orthopedic and Traumatology (DICHIRONS), University of Palermo, 90127 Palermo, Italy; salvatore.cassaro@community.unipa.it (S.C.); ludovico.lucenti@gmail.com (L.L.); lawrence.camarda@unipa.it (L.C.)
4. Department of Orthopaedics and Traumatology, Fondazione Policlinico Universitario A. Gemelli IRCCS, Università Cattolica del Sacro Cuore, 00168 Rome, Italy; giuseppe.rovere02@icatt.it
5. Department of Clinical Science and Translational Medicine, Section of Orthopaedics and Traumatology, University of Rome "Tor Vergata", 00133 Rome, Italy
6. Department of Orthopaedics and Traumatology, Ospedale San Giovanni Bosco di Torino-ASL Città di Torino, 10154 Turin, Italy
7. Department of Orthopaedic and Traumatology, Orthopaedic and Trauma Center, University of Turin, 10125 Turin, Italy; virginia.masoni@unito.it (V.M.); marcello.capella@alice.it (M.C.); srisitano@gmail.com (S.R.)
8. Department of Robotic and Minimally-Invasive Arthroplasty Surgery, Humanitas Gradenigo, 10153 Turin, Italy; luigisabatini.ort@gmail.com
* Correspondence: fortunato.giustra@gmail.com

**Abstract:** While anterior cruciate ligament reconstruction (ACL-R) has been considered the gold standard for ACL tears, renewed interest in ACL repair has emerged. This review aims to examine the current knowledge regarding ACL repair. A comprehensive literature search was conducted on the PubMed, Web of Science, Scopus, and Embase databases, focusing on the most recent studies up to January 2024. Arthroscopic ACL repair has several advantages, such as resulting in a natural ligament with proprioceptive properties, preservation, and donor graft comorbidity absence. Several ACL repair surgical procedures have been developed thanks to the advancement in new fixation devices. The current literature showed that when performed on a suitable patient with the appropriate lesion type, corresponding to the proximal third with good tissue quality, ACL repair leads to satisfactory outcomes. Despite the benefits of ACL repair with promising results, ACL-R remains the gold standard for ACL lesions. There is still a lack of literature analyzing long-term outcomes; large series with homogenous populations and types of lesions are lacking. Based on the current evidence, further research and higher-quality studies investigating ACL repair will be necessary.

**Keywords:** ACL; anterior cruciate ligament; repair; state of the art; review

## 1. Introduction

Anterior cruciate ligament (ACL) treatment is one of the hottest topics in the orthopedic field and is continuously evolving [1,2]. Recent epidemiological studies have highlighted ACL tears as among the most prevalent knee injuries, with an estimated annual incidence ranging from 100,000 to 200,000 cases in the United States alone [3–5]. Moreover, ACL tears disproportionately affect athletes participating in high-risk sports such as soccer, basketball, and football, contributing significantly to morbidity and imposing a substantial economic burden [4,6].

The diagnosis of an ACL tear is established through a combination of patient history, physical examination, and diagnostic imaging modalities [7,8]. Patients often present with

a history of a sudden twisting or pivoting motion, followed by a popping sensation in the knee joint, swelling, and instability. Physical examination may reveal findings such as a positive Lachman test, anterior drawer test, or pivot shift test, indicating ligament laxity and instability [3,4,6]. Diagnostic imaging, including magnetic resonance imaging (MRI), is commonly used to confirm the diagnosis, visualize the extent of the injury, and assess for associated damage to other structures within the knee joint, such as meniscal tears or cartilage injuries [6–8].

Discussing ACL management necessitates the dichotomous distinction between ACL repair and reconstruction (ACL-R) [1,2,9,10]. Open ACL repair was the main surgical strategy in the 20th century; despite promising early results, high medium- to long-term failure rates led to the switch to ACL-R [1,2,9–12]. In recent decades, renewed interest in ACL repair has developed, as evidenced by the increased number of publications in the literature [1,2,9–11]. Technology advancements associated with appropriate patient and lesion selection have resulted in promising ACL repair outcomes [1,2,9–11,13,14]. Pang et al., in their meta-analysis, highlighted that for proximal ACL ruptures, arthroscopic ACL repair was reported to have similar clinical outcomes compared with ACL-R [15]. ACL repair has several advantages, such as preservation of the natural ligament and its proprioceptive properties and donor graft comorbidity avoidance [1,2,9–11,14]. Furthermore, ACL-R remains an option in the case of an unsuccessful repair outcome [1,9,13].

The ACL anatomy has also been analyzed, revealing its healing ability, especially in the proximal part, due to rich vascularization [11,16]. However, several factors are still related to ACL repair failure, and the lesion location is of utmost importance [1,2,10,17].

Several ACL repair surgical techniques have been developed due to the advancement in new fixation devices. Direct repair with sutures alone, the suture anchor technique, bridge-enhanced repair (BEAR), dynamic intraligamentary stabilization (DIS), adjustable-loop cortical button device techniques, and internal brace augmentation procedures have all been described [1,2,18]. Moreover, additional procedures, such as the anterolateral ligament complex reconstruction (ALL-R) or lateral extra-articular tenodesis (LET), have been associated [1,19].

There are very few investigations regarding rehabilitation protocols, and most programs are extrapolated from the ACL-R literature [1,2,14,20].

This study aims to review the current knowledge on ACL repair. First, it proposes a historical overview describing the advantages and drawbacks of repair and a description of the healing potential of the ACL. Secondly, the indications and contraindications for repair and a summary of the surgical techniques and associated procedures are reported. Finally, an introduction to the rehabilitation protocol is proposed.

## 2. Search Strategy

Extensive research into the current evidence regarding ACL repair was conducted utilizing databases such as PubMed, Web of Science, Scopus, and Embase. All studies included in the analysis were exclusively in English, and a wide range of evidence levels were considered. No specific time constraints were applied, enabling a thorough exploration of the latest advancements in ACL repair, incorporating literature up to January 2024. This approach ensured a comprehensive overview of the field's progress. Various aspects of ACL repair were meticulously summarized and presented in separate paragraphs to provide a thorough understanding of the topic.

## 3. The Historical Issues, Advantages, and Disadvantages of ACL Repair

A historical analysis is essential to introduce the ACL repair technique [1,2,9–11]. Open ACL repair was traditionally the cornerstone of ACL treatment until the 1980s, when several studies reported a significantly high failure rate in medium- to long-term follow-ups [1,2,9–12,18]. For example, Feagin and Curl [12] documented successful ACL repair but with a decline at approximately five years of follow-up, while Kaplan et al. [21] described the open primary ACL repair for mid-substance tears as being an "unpredictable"

procedure with a 17% failure rate. For these reasons, surgeons shifted towards ACL-R, the gold standard nowadays for ACL lesions [1,2,9–11,18].

However, in 1991, Sherman et al. reported that patients with proximal avulsions and those with remaining good tissue quality achieved good outcomes following ACL repair [22]. Van der List et al., by retrospectively analyzing the historical literature, described tear location as having a key role in open primary ACL repair outcomes [11]; the authors reported better outcomes in studies presenting numerous proximal tears and excellent results in works with only proximal tears [11].

Recently, with new technology advancements in the arthroscopy field, increasing interest in ACL repair has been reported in the literature [1,13–15,18]. For example, Duong et al. [14] reported that in proximal ACL tears, ACL repair with internal brace augmentation achieved comparable patient-reported outcome measures (PROMs) and superior return-to-sport outcomes at an earlier time point than reconstruction.

Renewed interest in ACL repair is derived from the potential advantages of this procedure concerning ACL-R, such as native tissue preservation with proprioceptive properties and the fact that it is a less invasive procedure with no donor graft harvest morbidity [1,2,9–11,14]. Moreover, in the case of a failed repair, ACL-R is equivalent to primary ACL-R [1,9,13].

One crucial aspect investigated recently is the ACL's proprioceptive property [1,2,9–11,18]; the ACL presents mechanoreceptors with proprioceptive properties responsible for knee stability feedback [18,23,24]. For this reason, even in ACL-R, several authors suggested leaving ACL remnants to provide better knee stability and position sense [18,24]. A recent study by Cho et al. described improvements in proprioception in leaving ACL remnants compared to a standard demolitive technique [25]. Vermeijden et al., comparing ACL repair to ACL-R, described how patients forgot more about their operated knee in everyday life after undergoing the first procedure compared to the second one in a short- to mid-term follow-up [26].

## 4. Challenging the Dogma: The Healing Potential of the ACL

Historically, there has been the assumption that the ACL cannot heal [9,27]. However, recent studies have demonstrated that some ACL injuries can heal [2,9–11,27].

Scapinelli, analyzing ACL vascularization, described the middle genicular artery as the main reference vessel, with the distal end of the ligament supplied by branches of the inferior genicular artery [28]. The author reported major healing difficulties for the ACL compared to a posterior cruciate ligament (PCL), especially after mid-third interruptions, due to scarce vascular supply from the inferior genicular artery at the distal stump [28]. Toy et al. [29] and Petersen et al. [16] reported the same conclusions and described inhomogeneous vascularization with higher vessel numbers in the proximal part (Figure 1).

Apart from the vascularization pattern, some researchers tried to understand the reason for the ACL's poor healing response compared to other knee ligaments such as the medial collateral ligament (MCL) [2,27,30,31]. A mechanical instead of a biological reason was hypothesized, since the environments surrounding these two torn ligaments differ [2,30,31]. The ACL is surrounded by synovial fluid with degradative enzymes, whereas the MCL and all other extra-articular ligaments do not present the same pattern [1,2,30,31]. Thus, ACL and MCL cells could activate the healing response equally, but there is no mechanical bridge between the two stumps in the ACL [2,30,31]. These findings led to the hypothesis that the lack of a provisional scaffold between the two torn ACL ends was the key mechanism at the base of the healing failure [2,27,30,31].

Moreover, as reported by Ferretti et al., the distal stump could bend on the PCL due to the gravity force, where it would eventually attach, losing its function [27].

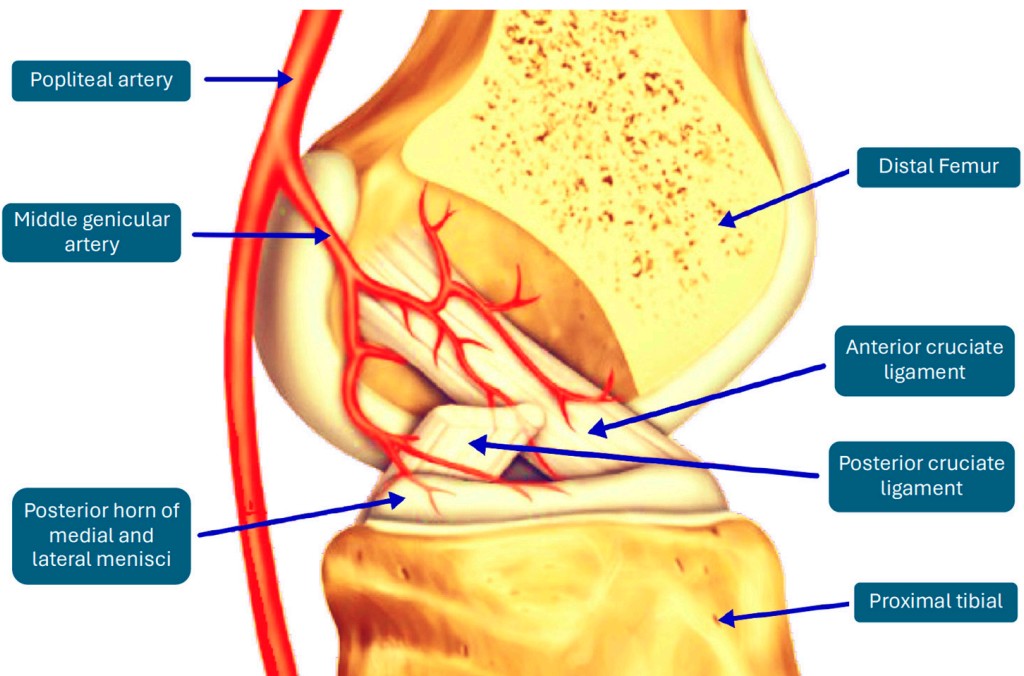

**Figure 1.** Vascular anatomy of the anterior cruciate ligament: a comprehensive illustration.

### 5. Indications and Contraindications for ACL Repair

In ACL repair, proper patient selection and tear pattern identification are fundamental since failures described in the past were mainly attributed to ACL repair in an un-selected target [1,2,10,17,22]. According to Sherman et al.'s classification, the ACL tear location could be identified and classified preoperatively in an MRI scan [22]. However, these lesions should be counterchecked intraoperatively since the intraoperative evaluation guides the final management [1,18].

The main aspects analyzed in the literature concerning ACL repair indications and contraindications are the tear location, the remaining tissue quality, the time from injury to repair, the patient's activity level, and the patient's age [1,2,17,22] (Table 1).

**Table 1.** Indications and contraindications for anterior cruciate ligament repair.

| Indications for ACL Repair | Contraindications to ACL Repair |
| --- | --- |
| Adult [1,13,22,32–35] | Adolescent [1,13,22,32–35] |
| Acute injuries [2,17,32] | Chronic tears [2,17,32] |
| Proximal ACL tears, femoral avulsion, type I-II Sherman [1,2,17,22] | Mid-substance tears [1,2,17,22] |
| | Poor tissue quality [1,2,17,22] |

- Authors agree that good tissue quality is needed as the baseline condition to perform the repair [1,2,17].
- Authors do not suggest performing the repair in patients with high activity level and those returning to competitive sports, with further investigations needed before adoption in sport medicine [1,2,13,22,32–35].

ACL—anterior cruciate ligament.

Tapasvi et al. identified the main indications for ACL repair: good tissue quality, an acute femoral avulsion type ACL injury (Type I Sherman), subsynovial ACL stretch/tear injury, partial ACL tear, and pediatric patients [17]. These indications were also described by Hughes et al. [2]. The literature agrees that worse outcomes are associated with mid-substance injuries, high-level sports athletes, chronic tears, and poor tissue condition [1,2,17,22].

Regarding tissue quality, Robinson et al. suggested intraoperatively probing the ACL stump in case of frayed fibers or mid-substance rupture before proceeding with ACL-R [1]. The most suitable lesions to undergo repair were proximal lesions [1,2,17,22].

Concerning the time from injury, surgery should be performed as early as possible because increased time negatively impacts the ACL ligament due to stump damages [1,17,32]. Van der List et al. demonstrated that surgery within four weeks has a higher probability of repair [32].

Age appears to be a relevant factor in ACL repair failure [1,22,32–35]. Ferreira et al. [34] reported a significantly higher level of failure in younger patients, in accordance with Vermeijden et al. [35], who found that patients younger than 21 reported a 37.0% failure rate, compared to 3.5% in those over 21 years. Higher failure rates in younger patients were observed after ACL-R, too, and this was supposed to be due to a potential higher activity level and an early return to sport [34,35]. However, adolescents still have a higher failure rate following ACL repair compared to ACL-R [1,33,34]; Ferreira et al. hypothesized that this result was related to biological differences in the ACL stump, with more frequent stump reabsorption in younger patients [34].

## 6. Surgical Techniques for ACL Repair

Several surgical techniques with different devices were developed for ACL repair [1,2,18]. The most frequent are summarized below.

### 6.1. Primary Repair with Suture Alone

The primary repair with a suture alone method, rooted in open suture-only fixation [2], entails the delicate insertion of loop sutures into the proximal ACL stump, anchoring them into precisely drilled holes within the ACL footprint of the femoral condyle [2]. Historically, this technique has been lauded for its simplicity and versatility, deemed applicable to a wide spectrum of injuries [2]. Moreover, its utilization served as a cornerstone in the early management of ACL ruptures, offering a direct approach to ligamentous repair.

However, it is essential to acknowledge the nuanced evolution of surgical techniques in addressing ACL injuries. While primary repair with suture alone may have been relegated to the annals of orthopedic history, its contributions to the field's understanding and treatment of ligamentous injuries remain noteworthy. Indeed, pioneering surgeons honed their skills through this method, laying the groundwork for subsequent advancements [2,18,36].

As advances in arthroscopic methodologies continue to redefine surgical norms, the legacy of earlier techniques underscores the iterative nature of medical progress. Nonetheless, the principles and insights gleaned from primary repair with suture alone persist, informing contemporary practices and contributing to the collective knowledge base of orthopedic surgery that has, over time, developed toward the use of arthroscopy procedures with suture anchors and augmentation [18,36].

### 6.2. Suture Anchors

Building upon the principles of the suture-alone technique, using suture anchors represents a pivotal advancement in ACL repair procedures. With this approach, the sutured ACL is affixed to the ACL footprint of the femoral condyle using anchors, providing a level of precision and strength unparalleled by previous methods [1,2,18,37] (Figure 2). This technique heralds a new era in ligament repair, offering surgeons enhanced control and stability in securing the ACL to its anatomical attachment points.

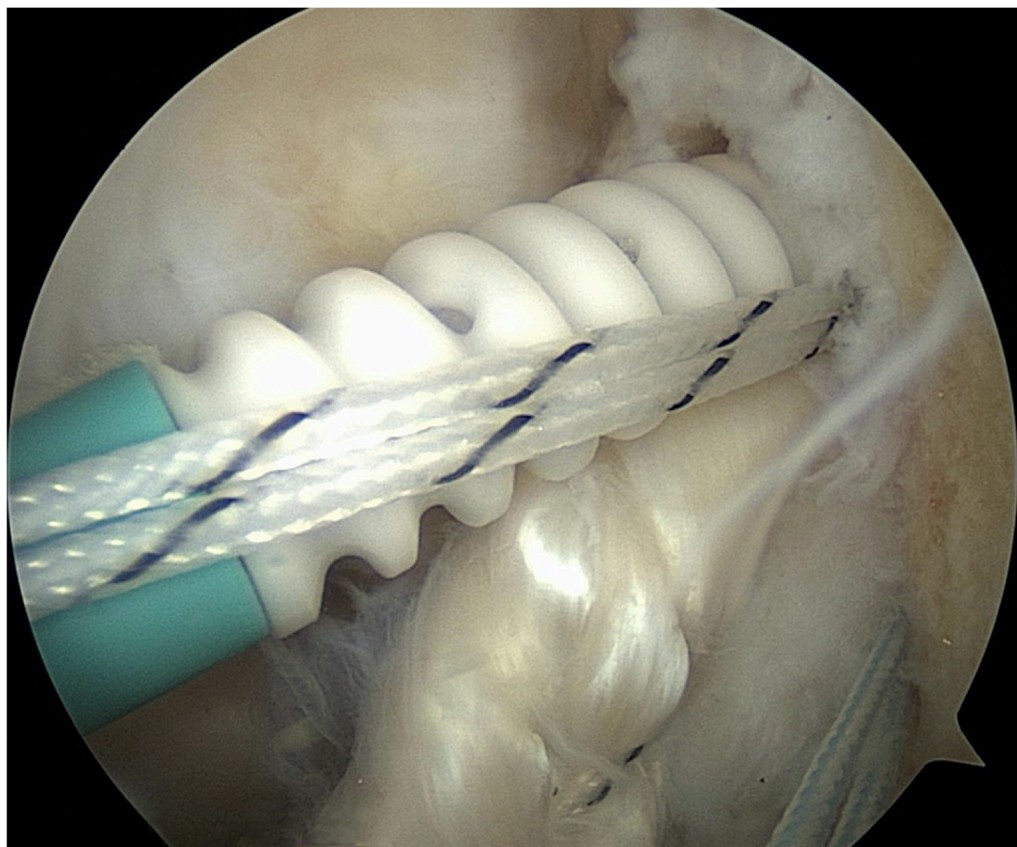

**Figure 2.** A suture anchor being deployed into the femur toward the anteromedial bundle origin to tension the anterior cruciate ligament remnant up to the wall (arthroscopic image of a left knee, viewed from the anterolateral portal, with the patient supine and the knee at 90 of flexion). The source is published from DiFelice et al. [37], under an agreement between Francesco Bosco and Elsevier. License Number: 5745371027644.

DiFelice et al. have underscored the efficacy of suture anchors, reporting favorable outcomes even at intermediate follow-up periods. Their research has demonstrated promising results, particularly in cases where the ACL tissue quality is robust and proximal avulsions are present [37]. Such findings underscore the importance of patient selection and the surgical technique in optimizing the outcomes of ACL repair procedures employing suture anchors [37].

Furthermore, the widespread adoption of suture anchors has facilitated the evolution of minimally invasive arthroscopic approaches, enabling surgeons to navigate complex anatomical structures more precisely. This paradigm shift has improved patient outcomes and streamlined the rehabilitation process, minimizing postoperative morbidity and expediting return to function [1,2].

As we continue to refine our understanding of ACL injuries and their management, the advent of suture anchors stands as a testament to the relentless pursuit of excellence in orthopedic surgery.

### 6.3. Dynamic Intraligamentary Stabilization (DIS)

Dynamic intraligamentary stabilization (DIS) differs from the previous system because it is a dynamic device [1,18,38,39]. While sutures supply a rigid fixation, this system can withstand cyclic loading during knee motion [18,38,39]. It consists of a spring–screw mechanism; the spring is inserted into the tibia, while a wire is passed through the torn ACL, and it is fixed on the femur with a flip button, maintaining the tibia in posterior translation during every degree of knee flexion and thus approximating the two ACL

stumps as close as possible [18,38–40] (Figure 3). Malahias et al. reported that more clinical studies investigated the DIS, reporting promising results [18]. This is in line with the systematic review by Ahmad et al., which grouped levels I, II, III, and IV clinical studies, suggesting a role for the use of DIS in ACL repair treatment in selected patients [38]. On the contrary, in a recent multicenter study, Senft et al. described a 16.3% re-operation rate after ACL DIS repair and an anteroposterior laxity $\geq$ 3 mm in 18.2% of patients [41].

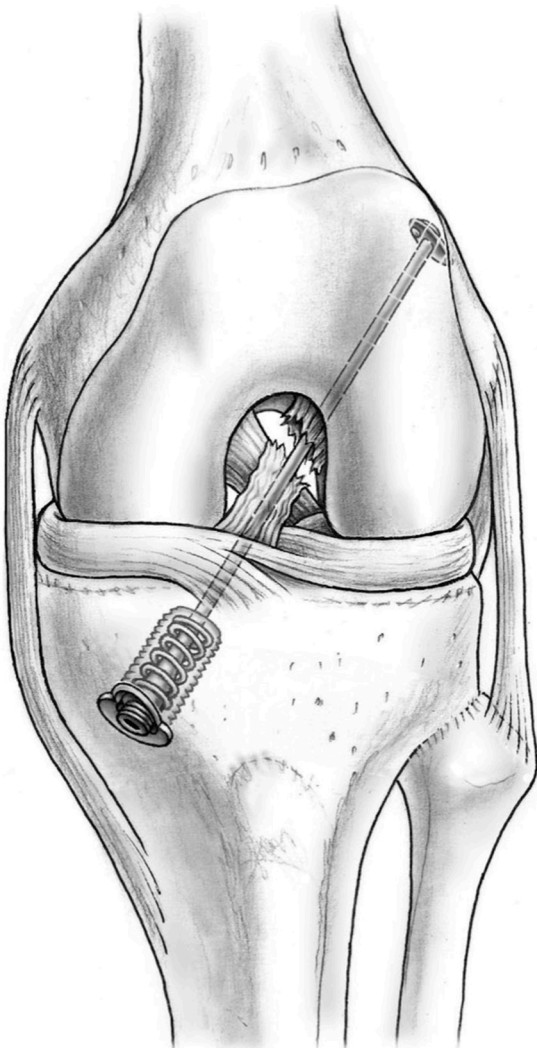

**Figure 3.** Dynamic screw–spring mechanism pushes the tibia into a posterior translation at every degree of flexion. The source is published from Eggli et al. [40] under a Creative Commons License.

### 6.4. Bridge-Enhanced ACL Repair (BEAR)

The BEAR technique represents a pioneering approach in the realm of ACL repair, integrating direct primary ACL repair with the application of an extracellular matrix scaffold [1,2,18,42,43]. This method stands out for its departure from conventional ligament repair practices, as it obviates the necessity of approximating ligament stumps. Instead, it capitalizes on the scaffold's inherent biological properties, serving as a conducive environment for the ACL stump to integrate and gradually replace the scaffold over time, facilitating tissue regeneration and functional recovery [1,2,18,42,43] (Figure 4).

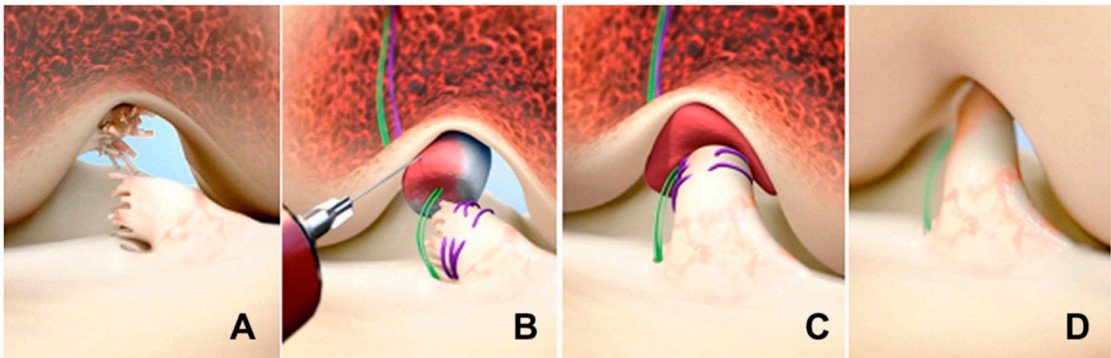

**Figure 4.** Stepwise demonstration of the bridge-enhanced anterior cruciate ligament repair (BEAR) technique using the scaffold. (**A**) The torn anterior cruciate ligament (ACL) tissue is preserved. A whipstitch of No. 2 absorbable suture (purple) is placed into the tibial stump of the ACL. Small tunnels (4 mm) are drilled in the femur and tibia, and a cortical button with two No. 2 nonabsorbable sutures (green sutures) and No. 2 absorbable sutures attached to it is passed through the femoral tunnel and engaged on the proximal femoral cortex. The nonabsorbable sutures are threaded through the BEAR scaffold and tibial tunnel and secured in place with an extracortical button. (**B**) The scaffold is then saturated with 5 to 10 mL of the patient's blood, and (**C**) the tibial stump is pulled up into the saturated scaffold. (**D**) The ends of the torn ACL then grow into the scaffold, which is gradually replaced by healing ligament tissue. The source is published from Murray et al. [42] under a Creative Commons License.

In a study conducted by Murray et al., the efficacy and potential enhancements of this innovative technique were investigated [42,43]. Notably, the incorporation of whole blood into the collagen-based scaffold was explored to augment the healing process, capitalizing on the regenerative potential of blood components such as growth factors and cytokines [42,43]. Results from this investigation revealed promising outcomes, with the BEAR technique yielding non-inferior PROMs and comparable anteroposterior knee laxity when juxtaposed with autograft ACL-R. Murray et al. also noted an intriguing finding: individuals subjected to BEAR exhibited superior hamstring muscle strength in comparison to those who underwent traditional ACL-R [42,43].

The incorporation of an extracellular matrix scaffold in conjunction with direct primary ACL repair opens new avenues in ACL treatment, heralding a shift towards biological augmentation and tissue regeneration strategies. Further exploration into the mechanistic underpinnings and long-term outcomes of the BEAR technique holds promise for advancing the field of orthopedic surgery and improving patient outcomes in ACL injury management.

*6.5. Adjustable-Loop Cortical Button Devices*

Primary ACL repair using the ACL repair TightRope® implant and FiberRing™ sutures represents a breakthrough approach to ligament repair, offering a promising alternative to traditional techniques [44,45].

The ACL repair TightRope® implant, characterized by high tensile strength and biocompatibility, secures the ACL in its original position to the ACL footprint of the femoral condyle, facilitating primary healing and biological incorporation, thus minimizing the need for autograft or allograft tissue. The implant is complemented by the FiberRing™ suture system, which provides additional reinforcement and stability to the repaired ligament, promoting optimal load distribution and biomechanical integrity. The integration of a FiberTape® suture within the ACL repair TightRope implant facilitates the InternalBrace™ technique, a revolutionary approach that reinforces the repaired ligament with additional support and stability. Combining the benefits of knotless fixation with the biomechanical advantages of FiberTape® sutures empowers surgeons to achieve optimal tensioning and

fixation, promoting accelerated healing and improved outcomes for patients undergoing ACL primary repair [Figure 5].

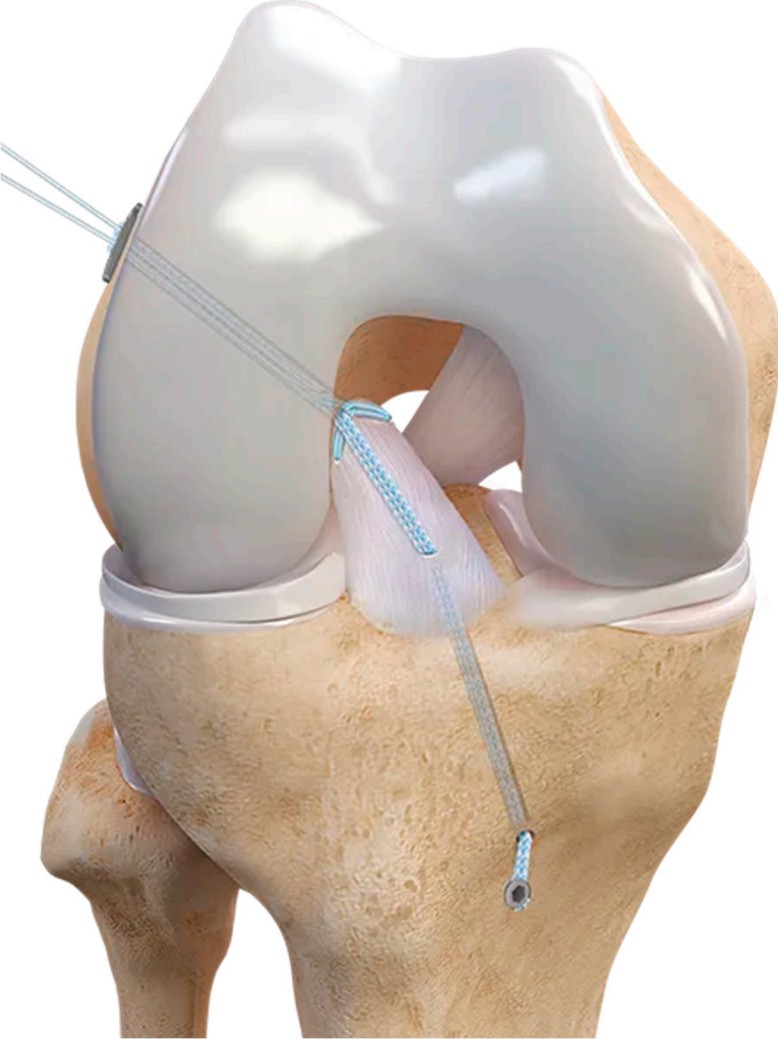

**Figure 5.** Schematic representation of the ACL repair TightRope® implant and FiberRing™ suture system in ACL repair surgery. The ACL repair TightRope® implant enables precise, incremental repair tensioning and retensioning, while the FiberRing™ suture facilitates simplified and reproducible suture passing and implant loading. The InternalBrace™ technique, which incorporates FiberTape® sutures within the TightRope implant for femoral fixation and BioComposite SwiveLock® anchors for tibial fixation, enables successful ACL repair.

Recent studies [44,45] have demonstrated promising outcomes with this technique, highlighting improved knee stability, reduced graft morbidity, and accelerated rehabilitation compared to traditional anterior cruciate ligament reconstruction methods. Additionally, preserving native ligamentous tissue and proprioceptive fibers may confer long-term benefits regarding joint kinematics and functional outcomes [44,45].

## 7. Augmentation Procedures with ACL Repair

Augmentation techniques have emerged as a procedure for enhancing ACL reconstruction success [1,14,18]; they were applied both in ACL-R [46–48] and ACL repair [1,14,18,19]. In the ACL repair setting, they follow the same principles and indications as in ACL-R [46,47].

The two most notable augmentation strategies were suture-tape augmentation [47] and laterally based procedures, either LET or ALL-R [46,49].

Moreover, in ALC repair, to promote the healing response, micro-perforations and micro-fracturings of the femoral ACL footprint have been performed [1,18,40,50].

### 7.1. Suture-Tape Augmentation

The suture-tape augmentation concept, as described by the InternalBrace™ technique integrating FiberTape® sutures within the TightRope implant for ACL repair, has revolutionized the landscape of ACL repair, offering a novel adjunctive technique to enhance the biomechanical properties of ligament reconstruction [44,45]. Augmenting the repaired ligament with a high-strength suture tape serves as a reinforcement mechanism, providing additional support and stability during the critical phases of healing and rehabilitation [1,18,47,51].

Recent advancements in surgical techniques and materials have propelled the suture-tape augmentation approach to the forefront of ACL preservation strategies. Notably, a comprehensive systematic review and meta-analysis have shed light on the efficacy of ACL repair with suture-tape augmentation, particularly in the context of proximal ruptures. This analysis revealed promising outcomes, with a failure rate of only 10.4% reported among patients undergoing ACL repair with an internal brace [51].

Moreover, the benefits of suture-tape augmentation extend beyond mere structural reinforcement. Studies have suggested that suture-tape augmentation may accelerate healing, reduce postoperative laxity, and enhance proprioceptive feedback, optimizing functional outcomes and facilitating an early return to activity [1,18,47].

As we refine our understanding of ACL injuries and explore innovative treatment modalities, suture-tape augmentation stands as a testament to the evolution of orthopedic surgery. Its integration into ACL repair protocols signifies a paradigm shift towards more nuanced and personalized approaches to maximize patient outcomes and minimize the burden of postoperative complications.

### 7.2. Laterally Based Procedures

ALL-R and LET have emerged as valuable adjuncts to ACL repair surgery, aiming to enhance rotational stability in the knee joint [46]. These procedures represent significant advancements in addressing the multifaceted nature of ACL injuries, particularly in cases where rotational instability remains a concern.

Monaco et al. have contributed to the field by presenting a specific technique that combines ACL repair with ALL suture augmentation, offering a comprehensive approach to addressing ligamentous laxity and rotational instability [19]. This technique underscores the importance of addressing both the primary ACL injury and potential deficiencies in the lateral structures of the knee to optimize overall joint stability and function.

Similarly, Fayard et al. proposed a hybrid approach involving ALL-R with a gracilis graft in conjunction with ACL repair, further emphasizing the synergistic benefits of addressing multiple ligamentous structures simultaneously [50]. By augmenting ACL repair with ALL-R, surgeons aim to restore the complex biomechanics of the knee joint and mitigate the risk of postoperative instability.

Recent research by Ferretti et al. has provided valuable insights into the comparative effectiveness of ACL repair combined with ALL repair versus ACL reconstruction combined with LET. Their findings suggest non-inferior clinical outcomes with ACL + ALL repair compared to ACL-R + LET, highlighting the potential of laterally based procedures in achieving satisfactory functional outcomes and patient satisfaction [52].

These studies [46,50,52] underscore the evolving landscape of ACL repair surgery, wherein a nuanced understanding of knee biomechanics and ligamentous anatomy informs the development of tailored treatment algorithms. By integrating laterally based procedures into ACL repair protocols, surgeons can comprehensively address the complex nature of knee instability, ultimately optimizing patient outcomes and reducing the risk of postoperative complications.

### 8. Rehabilitation Program and Return to Sport after ACL Repair

Postoperative rehabilitation programs are of utmost importance after ACL-R to reach successful outcomes [1,2]. They are mainly based on quadricep reinforcement, the early recovery of range of motion, and a return to specific training no sooner than 6 to 9 months postoperatively [1,2].

The literature agrees that specific rehabilitation protocols after ACL repair are lacking and are usually extrapolated by ACL-R [1–6]. However, as reported by Duong et al. [14] and by Robertson et al. [1], without graft harvest comorbidity, a faster and less painful recovery is expected, with quicker patient confidence in mobility. Ferretti et al. demonstrated that combined ACL + ALL repair resulted in a faster return to the pre-injury sport level than ACLR + LET [52].

However, the most recent systematic review regarding ACL repair rehabilitation highlights the lack of standardized rehabilitation protocols, with the need for future studies to investigate the most appropriate programs to return to sports [20].

### 9. Comparing ACL Repair and Reconstruction Techniques

In ACL repair versus reconstruction, it is crucial to consider the materials used for ACL-R, including autogenous and allogeneic tissues, and the associated complications [1,9]. Autogenous tissues, such as the patellar tendon and hamstring tendons, have been traditionally favored for ACL-R due to their biomechanical properties and potential for integration [9–11,13,30,31]. However, harvesting these tissues can lead to donor site morbidity, including pain, weakness, and scarring, which may impact patient recovery and function postoperatively. Allogeneic tissues, such as cadaveric grafts, offer an alternative to autogenous grafts but carry the risk of disease transmission, immune rejection, and slower graft incorporation [6,9–11,13,30,31].

In contrast, renewed interest in ACL repair stems from its potential advantages over reconstruction, such as native tissue preservation and reduced invasiveness. With ACL repair, there is no need for graft harvesting, minimizing donor site morbidity and preserving the ligament's natural proprioceptive properties. Additionally, failed ACL repair can be converted to ACL reconstruction without compromising outcomes, providing a safety net for patients requiring further intervention [11,14,22,47,48].

Despite these advantages, ACL repair has limitations [1,9–11,14]. The success of ACL repair depends on various factors, including tear location, tissue quality, and patient characteristics. Proximal avulsions and tears with remaining good tissue quality have shown promising outcomes with ACL repair, but mid-substance tears may be less amenable to this technique [27,30,31,33,34,47,48]. Furthermore, while ACL repair may offer improved proprioceptive outcomes compared to reconstruction, long-term studies are needed to assess the durability of these results and their impact on overall knee function [1,9–11,14,30,31,34,47].

In summary, the choice between ACL repair and reconstruction involves weighing the benefits and drawbacks of each approach, including considerations of tissue preservation, invasiveness, proprioceptive function, and potential complications [33,34]. While ACL repair offers certain advantages over reconstruction, including the preservation of native tissue and reduced donor site morbidity, careful patient selection and further research are necessary to optimize outcomes and ensure the long-term success of ACL treatment strategies [22–28,30,31,33,34,47].

### 10. Future Directions in ACL Repair

Significant improvements in ACL repair have been reported, mainly due to advances in surgical procedures and fixation devices [1,9–11,18], leading to satisfactory outcomes [13–15,52]. However, despite the promising ACL repair results, current evidence lacks high-quality, long-term studies. Therefore, ACL-R remains the gold standard for patients with ACL tears [2].

Future research, especially high-quality, long-term follow-up studies, is essential to explore the efficacy and applicability of this technique comprehensively. Moreover,

continued advances in this field may reveal more indications about the suitability of ACL repair for the appropriate patient and lesion type.

## 11. Conclusions

Renewed interest in ACL repair has emerged, and it is now accepted that ACL repair, when performed on a suitable patient with the appropriate injury type, has several advantages, such as natural ligament preservation with its proprioceptive properties and the avoidance of donor graft comorbidity.

Several ACL repair surgical techniques have been developed in recent years, and with advances in new fixation devices, arthroscopic ACL repair has reported promising results. Nevertheless, ACL-R remains the gold standard; ACL repair still requires further research and higher-quality studies.

**Author Contributions:** Conceptualization, F.B.; methodology, F.B. and S.C.; software, F.B.; validation, F.B., F.G. and G.R.; formal analysis, F.B. and F.G.; investigation, S.C., V.M. and F.G.; resources, F.B. and S.C.; data curation, F.B. and F.G.; writing—original draft preparation, F.B., S.C. and V.M.; writing—review and editing, V.M., F.B. and F.G.; visualization, F.B., G.R., F.G., M.C., S.R. and L.S.; supervision, F.B., F.G., G.R., L.L. and L.C.; project administration, F.B. All authors have read and agreed to the published version of the manuscript.

**Funding:** This research received no external funding.

**Institutional Review Board Statement:** Not applicable.

**Informed Consent Statement:** Not applicable.

**Data Availability Statement:** The data presented in this study are available in the article.

**Conflicts of Interest:** The authors declare no conflicts of interest.

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
