# Peer review of "Advancements in Anterior Cruciate Ligament Repair—Current State of the Art"

_2673-4095, doi:10.3390/surgeries5020022_

Round 1

Reviewer 1 Report

Comments and Suggestions for Authors

1. In the study inclusion criteria, please include the information on the language inclusion criteria, viz manuscripts from what all languages were considered in this review.

2. In section 3, please include a paragraph on the material used for ACL-R, viz autogenous and allogenous tissues, and the associated complications with using this treatment in the context of renewed interest ACL repair compared to reconstruction.

3. Please proofread the manuscript for errors, like line 122 “Authors try..” it should be “some researchers tried”…

4. “Authors” throughout the manuscript. It should be “authors”.

5. Line 211, PROMs is defined incorrectly.

6. Please remove section 10 “Strengths and Limitations’. This is not a primary research article.

7. “Institutional Review Board Statement”, “Informed Consent Statement”, “Data Availability Statement”. Why are these statements in the article? Not relevant and please proofread the whole article again before resubmission.

Comments on the Quality of English Language

Minor proofread needed.

Author Response

Thank you for your email with the reviewers’ comments.
We appreciate the helpful feedback from the reviewers on improving the quality and content of this manuscript.
The manuscript has been revised accordingly. Changes to the manuscript with a full-text rearrangement of the contents have been made.
We have addressed the specific reviewer concerns in a point-by-point manner below.
We hope that this revised manuscript is now suitable for publication and look forward to hearing from you.

Reviewer 2 Report

Comments and Suggestions for Authors

This manuscript aims to review the current knowledge on ACL repair. Overall, the manuscript is clearly written. However, the following statement might need clarification: “No time limitation was applied, focusing on the most recent literature, including up to January 2024.” How do you define the most recent literature?

Author Response

(The authors gave the same response as above.)

Reviewer 3 Report

Comments and Suggestions for Authors

This review presents intriguing insights that warrant publication. However, I have a few suggestions for improvement:

  1. In the introduction, consider incorporating statistics on the prevalence of ACL tear to provide context for readers.

  2. Include a brief discussion on the success rates of ACL repair procedures to enhance the understanding of potential outcomes.

  3. Explain how the diagnosis of ACL tear is typically established within the introduction for clarity.

  4. To augment the discussion on diagnostic tools, referencing relevant studies such as those available at:

    • https://pubmed.ncbi.nlm.nih.gov/34949491/
    • https://www.ncbi.nlm.nih.gov/pmc/articles/PMC3302044/ These studies highlight the utility of imaging modalities like MRI and ultrasound in detecting ACL tears.

By addressing these points, the review will offer a more comprehensive overview and contribute significantly to the field.

Author Response

(The authors gave the same response as above.)
